# A Review: Meridianins and Meridianins Derivatives

**DOI:** 10.3390/molecules27248714

**Published:** 2022-12-09

**Authors:** Linxia Xiao

**Affiliations:** School of Pharmacology, Jiangsu Vocational College of Medicine, Yancheng 224005, China; xiaolinxiaxlx@126.com

**Keywords:** meridianins, indole alkaloids, biological activities, pharmacological applications, pharmacokinetic characters, chemical synthesis, prospects

## Abstract

Meridianins are a family of indole alkaloids derived from Antarctic tunicates with extensive pharmacological activities. A series of meridianin derivatives had been synthesized by drug researchers. This article reviews the extraction and purification methods, biological activities and pharmacological applications, pharmacokinetic characters and chemical synthesis of meridianins and their derivatives. And prospects on discovering new bioactivities of meridianins and optimizing their structure for the improvement of the ADMET properties are provided.

## 1. Introduction

Natural products have always been an abundant source of varying chemicals that show multitudinous biological activities and have performed crucial functions in new drug research and development in multiple disease areas [1]. Marine creatures are among the most attractive resources of bioactive compounds [2]. One such example is the meridianins, which are a family of marine-derived alkaloids. Since meridianin A–E were first isolated from the Antarctic tunicate *Aplidium meridianum* in 1998 [3], a total of eight secondary metabolites (meridianin A–H) have been reported to be isolated and characterized so far [4]. Additionally, the *Aplidium meridianum*, meridianins have also been isolated from Antarctic tunicates *Aplidium falklandicum* and *Synoicum* sp. [5,6]. The basic structure of the meridianins is characterized by a brominated and/or hydroxylated indole framework linked to a 2-aminopyrimidine moiety at C-3 position (Figure 1). It is reported that natural products with an indole heterocycle usually possesses extensive bioactivity [7]. As indole alkaloids, the meridianins have displayed a myriad of pharmacological activities, such as inhibition of various protein kinases [8], anticancer activities [9], antimalarial activities [10], antituberculosis activities [11], anti-neurodegenerative activities [12], antibacterial activities [13], and have aroused a great deal of interest to researchers.

## 2. Extraction and Purification Methods

Three extraction approaches of meridianins are currently known (Table 1). One method was to extract the triturated tunicate with acetone thrice and sequentially partition against diethyl ether thrice and butanol once, which was the main reported method for extracting meridianins [5,14,15]. Using ethanol as a solvent to extract the triturated tunicate three times was another approach [8,16]. A third way was to extract the lyophilized tunicate with 1:1 dichloromethane/methanol thrice, subsequently partition between hexane and 95% aqueous methanol, and the condensed aqueous layer was sequentially partitioned between ethyl acetate and water to desalt and collect the ethyl acetate layer [4]. After evaporating the solvents under reduced pressure, the obtained dry residues were later used for target compound purification and chemical analysis.

To determine the presence of meridianins, thin layer chromatography on Merck Kieselgel plates using chloroform/methanol (8:2) as eluents, was used to screen the residues [17]. After reacting with CeSO_4_, a conspicuous yellowish UV-visible band at Rf 0.63 appeared in the residues composed of meridianins. The residues containing meridianins were further separated by molecular exclusion chromatography on Sephadex LH-20 columns by using chloroform/methanol 1:1 as solvent [15]. Alongside molecular exclusion chromatography, residues were fractionated by column-chromatography on silica gel using a petroleum ether/diethyl ether gradient or methanol/water 1:1 [3,18]. The obtained fractions were analyzed by 1H-NMR spectroscopic to determine whether each fraction was a pure compound or a mixture. Eluted fractions made up of a mixture were further chromatographed on reverse-phase semipreparative columns using HPLC techniques.

## 3. Biological Activities

### 3.1. Protein Kinase Inhibitory Potencies

Protein phosphorylation catalyzed by protein kinases, is implicated in all of the physiological processes for its function of driving signal transduction [19,20]. The pathogenesis of the majority of human diseases involves dysregulation of kinase activity [19]. Therefore, kinases have been extensively researched as drug targets in the twenty-first century [21]. Meridianins have been found to be a family of potent kinase inhibitors against 6 kinases (Table 2) [8] and inhibitory activities of meridianin E over the other 25 purified kinases have been further tested for it is the most active inhibitor against multiple kinases. Meridianin C also showed inhibitory activity towards Pim-1 kinase with an IC_50_ value of 1.0 μM and a family of its derivatives substituted at the 3-position and 5-position of the indole were synthesized with potent kinase inhibitory properties against Pim-1 and Pim-3 [22,23,24]. In addition, meridianin C derivatives were prepared as a GSK-3β inhibitor using a structure-based design and the appropriate insertion of compound into the ATP-binding pocket of GSK-3β might lead to a stronger kinase inhibitory activity based on molecular docking analysis [25]. At the same time, meridianin G derivatives modified at 5′-position were synthesized, and for multiple kinases (CK1δ/ε, GSK-3α/β, Dyrk1A, Erk2), their best inhibitory activities were sub-micromolar [26]. Giraud et al., also prepared a series of meridianin analogues, which were especially promising in the development of novel CLK1 and Dyrk1A kinases inhibitors [27]. Based on the above studies, meridianins could be taken as lead compounds for the synthesis of multiple kinase inhibitors.

### 3.2. Antiprotozoal and Antimicrobial Activities

According to literature reports, indole alkaloids have exhibited potential antimalarial activity [28,29,30]. As indole alkaloids, meridianin A was first reported to have potent antiplasmodial activity against *P. falciparum* (IC_50_ = 12 μM) in 2011 [10]. Subsequently, meridianin C and G were also found to possess significant antiplasmodial activity, and meridianin C displayed moderate antileishmanial activity against *Leishmania donovani* promastigotes with an IC_50_ of 64.9 μM [13]. Furthermore, a family of their derivatives were synthesized and evaluated for antimalarial activity, and the lowest IC_50_ value of them was 2.56 μM [11].

Besides antiprotozoal activities, meridianins also possess antimicrobial activities. Yadav et al., first demonstrated the antitubercular activity of meridianins [11]. The results of a microdilution assay revealed that the MIC values of meridianin C and G against a sensitive *M. tuberculosis* strain H37Rv were 111.1 and 304.8 μM, respectively. A crystal violet reporter assay indicated that meridianin D exhibited moderate biofilm inhibition activity against *M. smegmatis* and methicillin-resistant *S. aureus* with IC_50_ values of 21.5 and 87.4 μM, respectively [31,32]. Further experiments showed that meridianin D analogues increase colistin potency in polymyxin-resistant Gram-negative bacteria for their biofilm inhibition activities [33]. Hence, development of meridianin D analogues as adjuvants might be a promising strategy to combat the problem of antibiotic resistance. Moreover, meridianin C and a few derivatives displayed antifungal activity against *C. neoformans* [13]. Meanwhile, meridianin C also showed excellent anti-tobacco mosaic virus activity and broad-spectrum fungicidal activities [34]. As a consequence, it could serve as a lead compound to develop a fungicide for crop protection.

### 3.3. Chemical Defense Function

Deterrent secondary metabolites are one of the important components of defensive chemistry, which seems to be the first line of defense of most ascidians against predation [35,36,37]. Based on the Fisher’s exact test, Núñez-Pons and co-workers first indicated that the ether extracts of ascidian *A. falklandicum* led to remarkable deterrence against the predator sea star *O. Validus* and meridianins contained in the extracts turned out to be responsible for the protective effect [5]. According to further experiments carried out with isolated metabolites, the meridianins mixture displayed significant feeding repellence against sympatric predation at the natural concentrations and potent inhibition activity against an undisclosed sympatric marine bacterium [15]. Additionally, among multiple metabolites (including meridianin A–G, 5α(H)-cholestan-3-one, wax esters, rossinone B and glassponsine) obtained from Antarctic invertebrates, repellency levels of meridianin alkaloids were the highest [17].

### 3.4. Other Biological Activities

In addition to the abovementioned biological activities, meridianin A also reported to possess binding affinity towards central nervous system (CNS) receptors and transporters [10]. Secondary screening indicated that meridianin A exhibited measurable binding inhibition of radioligand to 5-HT2B with a Ki value of 150 nM and weakly inhibited binding of radioligand to 5-HT1A. At the same time, meridianin A also displayed binding inhibition of the radioligand to the dopamine active transporter. Furthermore, meridianin C was reported to reduce adipogenesis in 3T3-L1 adipocytes in a dose-dependent manner [38].

## 4. Pharmacological Applications

### 4.1. Anticancer Effect

Meridianins and meridianins derivatives showed anticancer activities against various cancer cell lines [27,39,40]. The anticancer activities of meridianins are listed in Table 3. Franco et al., indicated that meridianin B–E could suppress the growth of murine mammarian adenocarcinoma cell line LMM3 [3]. According to the evaluation of cell proliferation, meridianin A displayed cytotoxic activity against lung cancer cell line A549 [10]. Meridianin D exhibited weak antiproliferative activities toward several tumor cell lines based on the sulforhodamine B (SRB) assay [41]. An experiment showed that cyano meridianin D possessed strong cytotoxicity to breast carcinoma cell line MCF7 and cervix carcinoma cell line HeLa [9]. Additionally, high in vitro cytotoxicities toward breast carcinoma cell line MCF7 and human ovarian teratocarcinoma cell line PA1 were revealed for meridianin G derivatives [42]. Furthermore, meridianin analogues with the 2-aminopyrimidinyl ring replaced by pyrazolo [1, 5-α]pyrimidine ring displayed potent cytotoxic activities against human colorectal carcinoma cell line HCT-116 [43].

Previous studies showed that anticancer effects of meridianins and their derivatives might be regulated by multiple mechanisms. An article indicated that meridianin C could significantly suppress the growth of human tongue cancer cell line YD-10B and the antiproliferative function was mediated by micropinocytosis via down-regulation of DKK-3 [44]. Several studies found that meridianin C and its derivatives could inhibit the proliferation of three human leukemia cell lines (MV4-11, K562, and Jurkat) and the mechanism of action might involve pro-apoptosis via regulation of caspase-9, caspase-3, PARP, Mcl-1, Bcl-2, XIAP, eIF-2α and S6 molecules [22,23,45]. In addition, meridianin A, C, D and G showed weak antitumor activity against four human cancer cell lines: A549, DU14, HeLa and MDA-MB-231, in which JAK/STAT3 signaling is hyperactivated [46]. Whereas, the majority of their analogues displayed promising cytotoxicities toward the detected cell lines. According to western blotting assays, the most active analogue could down-regulate the phosphorylation levels of JAK1, JAK2, STAT3 and the protein expression levels of the downstream genes of STAT3 (c-Myc, Cyclin D1 and Bcl-XL), which demonstrated that the pro-apoptotic effects of the analogue might be mediated by regulation of the JAK/STAT3 signaling. All of the reported antitumor mechanisms of meridianins and their derivatives are presented in Figure 2.

### 4.2. Prevention of Alzheimer’s Disease (AD)

AD is the most common type of dementia whose pathologies are related to the existence of neurofibrillary tangles (NFT), mainly composed by aberrant phosphorylated tau protein [47,48,49]. GSK-3β, CK1δ, Dyrk1A and CLK1 have been recognized as the main protein kinases participated in tau phosphorylation [50,51]. Several articles demonstrated, in silico, that meridianins could bind to and inhibit the mentioned protein kinases, suggesting potential therapeutic application of meridianins in AD [12,52,53,54]. In fact, meridianin G, meridianin C and multiple derivatives displayed distinct Dyrk1A inhibition activity based on in vitro kinase inhibition assays and a meridianin C derivative also showed an apparent neuroprotective effect against neurotoxicity induced by glutamate [55]. Shaw et al., also demonstrated that meridianin derivatives with a 6-azaindole scaffold possessed high Dyrk1A binding activities [56]. Furthermore, in vitro assays indicated that meridianins also could inhibit the activity of GSK-3β and induce structural neuronal plasticity in primary cortical neurons [18]. In addition to inhibiting neural GSK-3β in vitro and in vivo, meridianins were able to ameliorate cognitive deficits and neuroinflammatory processes and induce structural synaptic plasticity in the 5xFAD mouse model of AD [57]. In conclusion, meridianins could serve as a potential agent for the treatment of neurodegenerative disorders.

### 4.3. Antimalarial Effect

Malaria, a devastating disease in developing countries, is caused by *Plasmodium* infection, mainly by *P. falciparum* [58,59]. Agarwal et al., first reported that meridianin analogs, with piperidinyl and aryl substituted at 2′-position and 6′-position, possessed potent in vitro inhibitory activity towards the malaria parasite *P. falciparum* [29]. Additionally, meridianins A, C, G and several derivatives of them were also shown to have moderate to strong antimalarial activity against *P. falciparum* [10,11,13], which indicated that meridianins represented attractive lead compounds to develop novel antimalarial agents.

### 4.4. Antitubercular Effect

Tuberculosis, caused by *M. tuberculosis*, remains the major cause of mortality in the world [60]. An experiment using a modified REMA method demonstrated that meridianin C and G displayed inhibitory activity against *M. tuberculosis* [11]. Several studies showed that meridianin D and its analogues exhibited antibiofilm activity against *M. smegmatis*, a vicarious bacterium for *M. tuberculosis*, via dispersing pre-formed biofilms and suppression of biofilm formation [31,33]. According to these results, there is potential for using meridianins as leads to the development of adjuvants against antibiotic tolerance.

### 4.5. Other Pharmacological Effects

As GSK-3β is a promising target for diabetes therapy, glucose uptake effects of meridianin C and its analogues were evaluated for their potent kinase inhibitory activity toward GSK-3β [25,61]. The experimental results indicated that multiple analogues of meridianin C displayed high glucose uptake, which suggested that these analogues might be new leads, with the potential to treat diabetes. Additionally, meridianin A and its derivatives possessed potential to treat CNS diseases due to their ability to bind to several CNS receptors [10].

## 5. Pharmacokinetic Study

As is well known, pharmacokinetic characters, composed of absorption, distribution, metabolism, excretion and toxicity (ADMET) are extremely important during drug discovery [62,63]. Hence, several studies analyzed the pharmacokinetic properties of meridianins and their derivatives in silico or in vivo [12,18,25].

Llorach-Pares et al. predicted 21 ADMET properties of meridianins and three derivatives by proprietary models of machine-learning and pkCSM [12]. The results of absorption properties indicated a possibly good oral and intestinal absorbance and poor skin permeability of the measured compounds (Table 4). Distribution predictors showed that blood-brain barrier (BBB) permeability of all compounds was poor. Depending on the results of cytochrome P450 interaction, all of the compounds could be metabolized and act as inhibitors for some isoforms of cytochrome P450. For excretion properties, there might be non-scavenging problems because none of them were zymolyte of organic cation transporter 2. Additionally, the analyzed compounds would be likely to have apparent toxicity, apart from meridianin A and E. Consistent with these results, the results of another two in silico pharmacokinetic studies also showed that all of the meridianins seemed to have good intestinal absorption and poor BBB permeability [18,53].

Furthermore, an in vivo metabolic study of meridianin C indicated that it was mainly presented as a prototype in plasma and its major metabolic route was phase I hydration and then suffering phase II conjunction metabolic pathways [64]. And an in vivo pharmacokinetic assay showed that a derivative of meridianin C possessed high oral bioavailability (47.4%), which was obviously improved compared with its lead compound [25].

Together, based on the acquired results, it is necessary to optimize the structure of meridianins in order to improve the ADMET properties and obtain compounds that can be used in clinic.

## 6. Chemical Synthesis

To date, several synthetic methods of meridianin analogues have been reported. Among them, there are mainly five widely used synthetic strategies: application of the Bredereck protocol [65,66], cross-coupling [67,68,69], conversion of 3-cyanoacetyl indole [9,70], alkenylation and condensation reaction of indoles [71] and indolization of nitro-soarenes [72].

The Bredereck protocol is more convenient to obtain meridianin C, D, G and their derivatives and suits for large-scale preparation. This protocol was achieved starting from appropriate indoles then preparation of the N-protected indoles and 3-acylindoles in the presence of aluminum chloride and acetyl chloride (Figure 1). The corresponding enaminones were obtained in the presence of dimethylformamide dimethylacetal in DMF. Finally, enaminones converted to meridianin alkaloids through the formation of an aminopyrimidine ring reaction and N-tosyl deprotection. This protocol is the most widely used synthesis scheme and most research groups currently use this method to synthesize meridianin derivatives. On the basis of Bredereck’s work, Sperry et al. used 5,6-dibromoindole-3-carbaldehyde as the starting material to obtain the key intermediate in excellent yield over three steps, then used the well-established Bredereck route to achieve meridianin F (Figure 1).

Jiang et al. [67] and Müller et al. [69] separately reported a palladium catalyzed cross-coupling reaction of 3-indolylboronic and pyrimidine moieties to prepare meridianin analogues. Jiang’s protocol was a traditional Suzuki coupling reaction between indolyboronic acid and chloroaminopyrimidine. Müller provided a concise one-pot reaction by the Masuda borylation-Suzuki coupling (MBSC) sequence, which was applied to the formation of meridianin C, D, F, and G successfully. Additionally, Karpov et al. [68] described a concise synthesis of meridianins and derivatives based upon a consecutive carbonylative coupling-cyclocondensation sequence by carbonylative alkynylation (Figure 2).

Radwan et al. [9] reported a facile synthesis of indolylpyrimidines, which was achieved by a cyanoacetyl side chain at the indole 3-position using a guanidine moiety for the construction of the aminopyrimidine ring (Figure 3). Rodrigues et al. [70] also provided a reaction of indole derivatives with cyanoacetic acid followed by treatment with DMF-DMA to obtain meridianin alkaloid derivatives. Compared with the above two schemes, this method requires higher requirements for reaction conditions; high temperature and strong acid may limit the further application of the reactions.

Yu et al. [71] reported a metal-free direct alkenylation reaction of indoles by using ac-id-mediated substitution reactions of aoxo ketene dithioacetals with indoles. A general procedure for condensation of these indolyl/ketene monothioacetals and guanidine nitrate led to meridianin derivatives successfully. An N-deprotection reaction with tBuOK/DMSO under an atmosphere of oxygen afforded meridianin derivatives in yields ranging 76–83% (Figure 4).

Penoni et al. [72] afforded a novel and atom-economical indolization process to obtain meridianin derivatives in moderate to good yields by thermal annulation of nitrosoarenes with 2-amino-4-ethynylpyrimidine and 2-chloro-4-ethynylpyrimidine, respectively (Figure 5).

## 7. Conclusions and Future Prospects

Meridianin A–F are a class of indole alkaloids derived from Antarctic tunicates *Aplidium meridianum*, *Aplidium falklandicum* and *Synoicum* sp. In this article, extraction and purification methods, biological activities and pharmacological applications, pharmacokinetic characters and the chemical synthesis of meridianins and their derivatives have been reviewed. Meridianins and derivatives possess varieties of biological properties, including protein kinases inhibition, [8] anticancer activities, [9] antimalarial activities, [10] anti-neurodegenerative activities [12] and so on. Among the biological properties, protein kinase inhibitory potencies are one of the most important activities of them, which are closely related to multiple other activities. Meridianins and derivatives are able to inhibit the activity of various protein kinases, such as GSK-3β, CK1, PKA and PKG [8]. A recent study shows that overexpression of GSK-3β stimulates podocyte senescence [73]. Additionally, several studies report that GSK-3β involves in activation of NLRP3 inflammasome [74,75,76]. As inhibitors of GSK-3β, meridianins might possess anti-aging functions and anti-inflammatory activities. Therefore, more attention should be paid to discover new bioactivities of meridianins. Furthermore, in consideration of the problems with meridianin’s pharmacokinetic characters, great efforts should be made to optimize their structure to improve the ADMET properties.

## Data Availability

Not applicable.

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
