# Peer review of "A Review: Meridianins and Meridianins Derivatives"

_molecules, 2022, doi:10.3390/molecules27248714_

Round 1

Reviewer 1 Report

This review summarizes the extraction and purification methods, biological activities and pharmacological applications, pharmacokinetic characters and chemical synthesis of meridianins and their derivatives. I think that many researchers are likely to interest these results, and this work is valuable for publication in this journal. However, several errors indicated in the PDF file should be corrected before publication.

Author Response

 Thanks. The errors indicated in the PDF file have been corrected.

Reviewer 2 Report

Meridianins are a class of very important alkaloids. The authors reviewed the extraction and purification methods, biological activities and pharmacological applications, pharmacokinetic characters and chemical synthesis of meridianins and their derivatives, which have certain significance for drug development. Throughout the whole paper, there are many small problems that need to be further improved by the author.

 1. The font of L17-L32 is different from the body font at the end. It may be bolded. Please check it by the author.

 2. When introducing the work of others, you should describe the difficulties and advantages of the work to achieve good results, rather than simply describing what the results of the other person's work were. Otherwise, such a statement would not be interesting to the reader.

Especially chapters 4 and 5.

 3. Chapters 3, 4, and 5 should have corresponding diagrams to aid the visualization of the work (if feasible), such as the chemical synthesis diagram done by ChemDraw (Chapter 6, L230).

 4. The arrows in Scheme 2 should indicate the specific work done by which team and the detailed conditions of response so that the reader can see the details of the work at a glance.

In addition, please rearrange the conditions of CuI and Pd(PhP3)3Cl2.

 5. In particular, the synthesis of Meridianin is very rich, and this paper has missed many groups' synthesis of Meridianin. Please refer to the following literature for detailed retrieval by Web of Science.

Such as:

Tetrahedron Letters, 2011, 52, 4537-4538.

Andrea Penoni Tetrahedron 2010, 66, 1280-1288.

Ligia M. Rodrigues, doi.org/10.3998/ark.5550190.0012.b11

Zhengkun Yu Angew. Chem. Int. Ed. 2009, 48, 2929-2933

 6. When commenting on a group's work in the body, do not use your full name. Use your last name instead. Such as Thomas J. J. Müller (P7, L246), Mohamed A. A. Radwan (P7, L257). Please check the whole article.

 7. There are some language grammar problems in the manuscript. Ask the author to find someone with native English background to polish it.

 Author Response

Thank you very much for giving me a chance to revise my manuscript entitled A Review: Meridianins and Meridianins Derivatives(Manuscript ID: molecules- 2065895). I would thank you very much for your constructive comments. According your comments, herein I would revise my manuscript point by point and make revisions as following.

  1. The font of L17-L32 is different from the body font at the end. It may be bolded. Please check it by the author.

 Answer: Thanks. The font of L17-L32 has been checked. The font of L17-L32 is the same as the body font at the end.

  1. When introducing the work of others, you should describe the difficulties and advantages of the work to achieve good results, rather than simply describing what the results of the other person's work were. Otherwise, such a statement would not be interesting to the reader. Especially chapters 4 and 5.

Answer: Thanks. Your advice is of great value. However, authors usually do not illustrate the difficulties and advantages of the work to achieve good results. So, they are not described in the manuscript.

  1. Chapters 3, 4, and 5 should have corresponding diagrams to aid the visualization of the work (if feasible), such as the chemical synthesis diagram done by ChemDraw (Chapter 6, L230).

Answer: Thanks. To aid the visualization of the work, a corresponding diagram has been added in the chapter 5.

  1. The arrows in Scheme 2 should indicate the specific work done by which team and the detailed conditions of response so that the reader can see the details of the work at a glance. In addition, please rearrange the conditions of CuI and Pd(PhP3)3Cl2.

Answer: Thanks. The name of team and the detailed conditions have been added to the arrows in Scheme 2. And the conditions of CuI and Pd(PhP3)3Cl2 have been rearranged.

  1. In particular, the synthesis of Meridianin is very rich, and this paper has missed many groups' synthesis of Meridianin. Please refer to the following literature for detailed retrieval by Web of Science. Such as: Tetrahedron Letters, 2011, 52, 4537-4538. Andrea Penoni Tetrahedron 2010, 66, 1280-1288. Ligia M. Rodrigues, doi.org/10.3998/ark.5550190.0012.b11. Zhengkun Yu Angew. Chem. Int. Ed. 2009, 48, 2929-2933.

Answer: Thanks. Referring to the literature provided by you, two synthesis methods of Meridianin have been added in the chapter 6.

  1. When commenting on a group's work in the body, do not use your full name. Use your last name instead. Such as Thomas J. J. Müller (P7, L246), Mohamed A. A. Radwan (P7, L257). Please check the whole article.

Answer: Thanks. The non-standard name formats have been modified and the whole article has been checked.  

7. There are some language grammar problems in the manuscript. Ask the author to find someone with native English background to polish it.

Answer: Thanks. The language grammar problems in the manuscript have been corrected.

Reviewer 3 Report

The author summarized in this review the advances of meridianin alkaloids, including isolation methods, biological and pharmacological studies, pharmacokinetic characters and chemical synthesis. The organization and the English of manuscript are good. I would like to recommend this manuscript be published in Molecules after the revision of the followings:

 1. The contents of Section 4.3 and 4.4 are somewhat duplicated with those of Section 3.2.

 2. Scheme 1: tos-Cl → Tos-Cl

 3. Please carefully check the citation of Refs, for examples:

  Ref 10: Maiquan → Maignan; 3 authors was lost in the author list

  Ref 67: Müeller, T.J. → Müeller, T.J.J.

 Author Response

Thank you very much for giving me a chance to revise my manuscript entitled A Review: Meridianins and Meridianins Derivatives(Manuscript ID: molecules- 2065895). I would thank you very much for your constructive comments. According to your comments, herein I would revise my manuscript point by point and make revisions as following.

  1. The contents of Section 4.3 and 4.4 are somewhat duplicated with those of Section 3.2.

Answer: Thanks. The repeatability has been minimized. However, since biological activity is the basis of pharmacological application, duplication between them is inevitable.

  1. Scheme 1: tos-Cl → Tos-Cl

Answer: Thanks. The error you pointed out has been corrected.

  1. Please carefully check the citation of Refs, for examples:

  Ref 10: Maiquan → Maignan; 3 authors was lost in the author list

  Ref 67: Müeller, T.J. → Müeller, T.J.J.

Answer: Thanks. The citation of Refs has been carefully checked and corrected.

Round 2

Reviewer 2 Report

No